# Effect of Acute Dietary Nitrate Supplementation on the Venous Vascular Response to Static Exercise in Healthy Young Adults

**DOI:** 10.3390/nu14214464

**Published:** 2022-10-24

**Authors:** Anna Oue, Yasuhiro Iimura, Akiho Shinagawa, Yuichi Miyakoshi, Masako Ota

**Affiliations:** 1Faculty of Food and Nutritional Sciences, Toyo University, Gunma 374-0193, Japan; 2Graduate School of Food and Nutritional Sciences, Toyo University, Gunma 374-0193, Japan

**Keywords:** beetroot juice, exercise, nitric oxide, sympathoexcitation, venoconstriction

## Abstract

The purpose of this study was to test the hypothesis that acute intake of inorganic nitrate (NO_3_^−^) via supplementation would attenuate the venoconstriction and pressor response to exercise. Sixteen healthy young adults were assigned in a randomized crossover design to receive beetroot juice (BRJ) or an NO_3_^−^-depleted control beverage (prune juice: CON). Two hours after consuming the allocated beverage, participants rested in the supine position. Following the baseline period of 4 min, static handgrip exercise of the left hand was performed at 30% of the maximal voluntary contraction for 2 min. Mean arterial pressure (MAP) and heart rate (HR) were measured. Changes in venous volume in the right forearm and right calf were also measured using venous occlusion plethysmography while cuffs on the upper arm and thigh were inflated constantly to 30–40 mmHg. The plasma NO_3_^−^ concentration was elevated with BRJ intake (*p* < 0.05). Exercise increased MAP and HR and decreased venous volume in the forearm and calf, but there were no differences between CON and BRJ. Thus, these findings suggest that acute BRJ intake does not alter the sympathetic venoconstriction in the non-exercising limbs and MAP response to exercise in healthy young adults, despite the enhanced activity of nitric oxide.

## 1. Introduction

Dietary nitrate (NO_3_^−^) supplementation with beetroot juice (BRJ) reduces resting blood pressure (BP) in normotensive and hypertensive populations [1,2,3,4,5,6]. The hypotensive effect of BRJ is likely to be due to both peripheral and central factors. The peripheral factor underlying the hypotensive effect might be related to the vasodilatory impact of the increase in nitric oxide (NO) bioavailability induced by the stepwise reduction of dietary nitrate (NO_3_^−^) to nitrite (NO_2_^−^) and subsequently to NO (i.e., the NO_3_^−^ → NO_2_^−^ → NO pathway) [4,7,8]. In addition to the role of NO as a vasodilator, the increase in NO availability related to the NO_3_^−^ → NO_2_^−^ → NO pathway may also alter efferent sympathetic outflow. Indeed, Notay et al. [9] recently reported that acute dietary NO_3_^−^ supplementation with BRJ decreased muscle sympathetic nerve activity (MSNA) at rest and blunted the MSNA response to sympathoexcitation via static handgrip exercise in young adults.

Veins have high distensibility and contain approximately 60–70% of the total blood volume at rest [10]. Venous tone is controlled by the sympathetic nervous system and changes with physiological stress (e.g., exercise), thereby leading to alterations in venous volume and/or compliance and contributing to the control of circulatory responses (e.g., BP and cardiac output) [10]. For example, venoconstriction and/or decreased venous compliance occurs sympathetically in the non-exercising limb during exercise [11,12,13,14,15,16]. In addition to sympathetic control, NO is also an important signaling molecule that contributes to the modulation of venous tone [17,18]. Moreover, in our recent study [19], it is suggested that the increased bioavailability of vasodilator NO associated with dietary NO_3_^−^ supplementation with BRJ could contribute to the control of the peripheral vascular tone in not only arteries but also veins under resting conditions, which may perhaps, in part, be attributable to the hypotensive effect of BRJ. However, the effect of BRJ intake on the venous vascular response to exercise-induced sympathoexcitation has not been investigated. Aging and physiological inactivity might cause stiffness of the veins [20,21], which could be a factor in the pathogenesis of hypertension [22]. Considering these findings, it is also important to understand the hypotensive effect of BRJ from the perspective of the venous vascular system.

Therefore, the purpose of this study was to investigate the effect of acute dietary NO_3_^−^ supplementation with BRJ on venous vascular and circulatory responses to static handgrip exercise in humans. Because the scientific evidence indicates that the NO_3_^−^ → NO_2_^−^ → NO pathway-related increase in NO bioavailability in response to acute dietary NO_3_^−^ supplementation with BRJ causes vasodilation [4,7,8] and attenuates the MSNA response to static handgrip exercise [9], we hypothesized that acute intake of BRJ would attenuate the venoconstriction and BP response to exercise.

## 2. Materials and Methods

### 2.1. Participants

Sixteen healthy individuals (10 men, 6 women, 22.3 ± 1.4 years, 167.4 ± 9.3 cm, 64.7 ± 13.9 kg) volunteered for this study. None of the women were using oral contraceptives and all were in the self-reported follicular phase (3–10 days after the onset of menstruation) during the experiments. This study was approved by the Human Ethics Committee of Toyo University (TU2019-018-TU2020-H-019) and was conducted in accordance with the Declaration of Helsinki. The purpose, procedure, and risks of the study were explained to the participants, and their written and verbal informed consent was obtained. Throughout the study, the participants were instructed to avoid vigorous exercise, caffeine, and alcohol for 24 h before each visit. For 3 days prior to the main experimental protocol and blood sampling protocol, participants were asked to refrain from high NO_3_^−^ foods (e.g., green leafy vegetables and traditional Japanese foods) [23,24]. In addition, because oral bacteria are involved in reducing NO_3_^−^ to NO_2_^−^ in vivo [25], the participants were asked to abstain from using mouthwash.

### 2.2. Experimental Design

Participants visited the laboratory on five occasions. During the first visit, they performed maximal voluntary contraction (MVC) of the left hand using a handgrip dynamometer to determine their 30% MVC. In addition, they were familiarized with the experimental procedure and equipment. Main experiments were carried out at the second and third visits, and blood samplings were performed at the fourth and fifth visits. For the main experiments and blood samplings, participants were assigned in a randomized crossover design to consume BRJ (Beet It^®^; James White Drinks, Ipswich, UK; 140 mL/day, containing ~8 mmol NO_3_^−^) or placebo control beverage (CON) consisting of prune juice (Sunsweet^®^; POKKA SAPPORO Food & Beverage Ltd., Nagoya, Japan; 166 mL/day; <0.01 mmol NO_3_^−^). We selected prune juice as the placebo beverage because the NO_3_^−^ in prune juice is at a negligible level and the carbohydrate and fiber contents of prune juice are similar to those of BRJ [26,27]. The amount (166 mL/day) of prune juice was calculated to match the energy contained in 140 mL/day of BRJ. A washout period of at least 7 days separated each supplementation period for both the main experiments and the blood sampling.

### 2.3. Protocol of the Main Experiment

After arrival at the laboratory, participants were instructed to consume BRJ or CON. Two hours later, all participants rested in the supine position for 20 min before data acquisition in an air-conditioned room (26.4 °C ± 0.5 °C). The main experiment comprised two protocols: (1) measurement of the changes in venous volume in the non-exercising limbs during the static handgrip exercise, and (2) measurement of the circulatory parameters during static handgrip exercise. In protocol 1, following the pre-exercise baseline period for 4 min, the static handgrip exercise of the left hand was performed at 30% MVC for 2 min. The cuffs on the right wrist and the right ankle were inflated to 200–220 mmHg at the same time as the start of the baseline period, and the cuffs on the right upper arm and the right thigh were inflated to 30–40 mmHg from the first minute of the baseline period. All cuff inflations were maintained until the end of the exercise. Throughout protocol 1, the changes in venous volume in the non-exercising right forearm and right calf were measured using venous occlusion plethysmography (Hokanson, EC6; D. E. Hokanson, Bellevue, WA, USA). In protocol 2 as well as in protocol 1, static handgrip exercise of the left hand at 30% MVC for 2 min was performed following the 4-min baseline period. Throughout protocol 2, the circulatory parameters were measured in the right middle finger using Finapres NOVA (Finapres Medical Systems BV, Enschede, The Netherlands). Protocol 1 and protocol 2 were performed in random order, and rest periods of at least 20 min were allowed between protocols 1 and 2. In addition, for all protocols, participants controlled their respiratory rate at 10 or 15 breaths per minute, guided by a metronome.

### 2.4. Blood Sampling Protocol

To measure the plasma NO_3_^−^ concentration, venous blood samples were drawn from an antecubital vein in a seated position before and 2 h after BRJ or CON consumption on a different day from the main experimental protocols.

### 2.5. Measurements

Systolic blood pressure (SBP), diastolic blood pressure (DBP), and mean arterial pressure (MAP) were measured noninvasively and continuously from the right middle finger using Finapres NOVA, which was calibrated with the right upper arm cuff and height adjustment. In addition, heart rate (HR) was determined from the BP waveform using the Modelflow software program (Finapres Medical Systems BV, Enschede, The Netherlands).

To assess venoconstriction in the non-exercising forearm and calf during static handgrip exercise of the left hand, the changes in volume in the right forearm and right calf were measured. Inflatable cuffs were wrapped around the right wrist, right upper arm, right ankle, and right thigh, and strain gauges were placed on the sites of maximal thickness in the forearm and calf. Throughout the protocol, the wrist and ankle cuffs were inflated to 200–220 mmHg to arrest the blood circulation of the hand and foot, which have arteriovenous anastomoses, because we wanted to investigate the venoconstriction in the forearm and calf. One minute after the cuff inflation of the wrist and ankle, the cuffs of the upper arm and thigh were inflated to 30–40 mmHg for 3 min, and the volume in the forearm and calf increased until it approached an asymptote, and then static handgrip exercise of the left hand was performed for 2 min. Throughout the protocol (total time, 6 min), the change in volume in the right forearm and the right calf was measured using venous occlusion plethysmography. This method was adopted because when the intravascular pressure of the conduit vein in the limb is maintained at a constant level, the decrease in the limb volume reflects the venoconstriction or the elevated venous vascular tone [15,16]. In addition, the cuff inflation pressure can be considered equivalent to the intravascular pressure of the conduit vein in the limb [28].

Venous blood samples were immediately mixed with EDTA and centrifuged at 3000× *g* rpm for 10 min. Plasma was placed in microcentrifuge tubes and frozen for the subsequent analysis of plasma NO_3_^−^ and NO_2_^−^ concentrations [29]. The NO_3_^−^ in a sample is reduced by a cadmium column to NO_2_^−^, which reacts with a Griess reagent to form a purple azo dye. The NO_3_^−^ in a sample needs no reaction and thus reacts with the Griess reagent when the cadmium column is bypassed, as well as when it is used. The dye was developed in a 60 °C water bath, the sample was cooled by a 0 °C water bath, and its absorbance at 546 nm was detected using a flow-through UV-Vis spectrophotometer (V-750, JASCO Corporation, Tokyo, Japan). Because the plasma NO_2_^−^ levels were very low and below the quantifiable limit (1 µM) in all participants, only NO_3_^−^ concentration data are presented in this study.

### 2.6. Data Analysis and Statistics

Data are expressed as the mean ± standard deviation. A priori sample size calculation estimated a required sample of 12 participants, assuming a change in venous compliance of 0.031 ± 0.027 mL/dL of tissue/mmHg [30] in a crossover trial with an assigned α of 0.05 and β of 0.2. Pre-exercise baseline values of SBP, DBP, MAP, and HR were defined as the mean value obtained from 1 to 3 min of the baseline period. In addition, pre-exercise baseline values of the forearm and calf venous volume were defined as the average of the last 10 s of the 3-min cuff inflation to 30–40 mmHg. The SBP, DBP, MAP, HR, and limb venous volume values during exercise at 2 min were obtained as the average of the last 1 min of exercise. Relative increases in SBP, DBP, MAP, and HR and relative decreases in venous volume in the forearm and calf with exercise at 2 min from the pre-exercise baseline were calculated. The relative decrease in venous volume was used as the index of venoconstriction.

To compare the changes in SBP, DBP, MAP, HR, and forearm venous volume and calf venous volume with exercise between CON and BRJ, a two-way analysis of variance (ANOVA) with repeated measurements (condition × time) was applied. If the main effect of condition (CON and BRJ), that of time (pre-exercise and during exercise), and/or an interaction effect were detected, post hoc analysis using a paired *t*-test was performed. To compare the plasma NO_3_^−^ concentration before and after BRJ or CON supplementation and the plasma NO_3_^−^ concentration between BRJ and CON, a paired *t*-test was used. In addition, we calculated the differences in the absolute MAP at 2 min of exercise between CON and BRJ, in the MAP elevation with exercise from pre-exercise (pressor response to exercise) between CON and BRJ, and in the relative decreases in venous volume in the limbs with exercise between BRJ and CON. Using these values, Spearman’s rank correlation coefficients were calculated to examine the relationships between venoconstriction and MAP responses to exercise with acute ingestion of BRJ. Statistical significance was set at *p* < 0.05. All statistical analyses were performed using SPSS version 27 (IBM Corp., Armonk, NY, USA).

## 3. Results

Figure 1 shows the plasma NO_3_^−^ concentration before and after intake of CON and BRJ. The plasma NO_3_^−^ concentration increased in all participants after BRJ ingestion (before, 15 ± 6 µM; after, 574 ± 120 µM; *p* < 0.05). However, there was no significant change in the plasma NO_3_^−^ concentration after CON ingestion (before, 14 ± 3 µM; after, 14 ± 3 µM). Furthermore, there was a significant difference in the plasma NO_3_^−^ concentration between CON and BRJ after their intake (*p* < 0.05).

Static handgrip exercise under both CON and BRJ conditions caused similar increases in SBP, DBP, MAP, and HR (Table 1). ANOVA indicated a significant time effect (all *p* < 0.01), and post hoc testing revealed significant differences in SBP, DBP, MAP, and HR between pre-exercise and during exercise (all *p* < 0.05). However, SBP, DBP, MAP, and HR at pre-exercise or during exercise were similar for CON and BRJ. In addition, the degrees of increases in these parameters during exercise from pre-exercise did not differ between CON and BRJ (Figure 2A–D). In contrast, static handgrip exercise under both CON and BRJ conditions induced similar decreases in venous volume in the non-exercising forearm and calf (Table 1). ANOVA indicated a significant time effect (all *p* < 0.01), and post hoc testing showed significant differences in these parameters between pre-exercise and during static handgrip exercise (all *p* < 0.05). This result meant that sympathetic venoconstriction was obtained during exercise. However, the venous volumes in the forearm and calf pre-exercise or during exercise were similar for CON and BRJ. In addition, the degrees of the decreases in the venous volume (venoconstriction) in the forearm and calf during exercise from pre-exercise did not differ between CON and BRJ (Figure 2E,F)

In the individual data, the difference in the decrease in forearm venous volume (venoconstriction) during exercise between BRJ and CON had no significant relationship with the MAP response to exercise (Figure 3A,B). In contrast, a smaller venoconstriction in the calf during exercise with BRJ rather than CON was significantly associated with a lower MAP elevation with exercise with BRJ but not with absolute MAP at 2 min of exercise (Figure 3C,D).

## 4. Discussion

The new findings in our study are (1) that the increases in SBP, DBP, MAP, and HR and decreases in the venous volume in the non-exercising forearm and calf during static handgrip exercise did not differ between CON and BRJ, despite the elevated plasma NO_3_^−^ concentration after BRJ ingestion, and (2) that, in the individual data, a greater attenuation in the decrease in calf venous volume with exercise after BRJ ingestion was significantly associated with a lower MAP elevation with exercise after BRJ ingestion but not with the absolute MAP at 2 min of exercise. These results suggest that the enhanced NO bioavailability induced by BRJ supplementation does not alter sympathetic venoconstriction in non-exercising limbs or the pressor response to exercise in healthy young adults. In addition, although BRJ does not cause a decrease in venoconstriction in all subjects, when the attenuation of venoconstriction in the calf during static handgrip exercise after BRJ ingestion is obtained, the degree of the MAP pressor response to exercise could be reduced to a degree dependent on the attenuation of venoconstriction.

In the present study, the venous volume in the non-exercising forearm and calf decreased with static handgrip exercise under both CON and BRJ conditions, and these decreases in the venous volume did not differ between the two groups (Table 1 and Figure 2E,F), despite the increased plasma NO_3_^−^ concentration after BRJ intake. These results suggest that the NO bioavailability increase induced by acute BRJ ingestion does not attenuate the venoconstriction in the non-exercising limbs during exercise, which does not support our hypothesis. This is the first study to investigate the effect of BRJ on the venous vascular response to exercise. Some studies have investigated the effect of NO on the arterial vascular response. For example, sympathetic vasoconstriction in human and animal preparations is inhibited by NO derived from both endothelial nitric oxide synthase [31] and neuronal nitric oxide synthase [31]. In contrast, several studies have reported that the NO bioavailability associated with dietary NO_3_^−^ supplementation (e.g., BRJ) does not alter sympathetic vasoconstrictor responsiveness to exercise [32,33]. Despite differences between arterial and venous vessels and between active and inactive limbs in the previous and present studies, our findings and those of the other studies suggest that exogenous NO augmentation via the NO_3_^−^ → NO_2_^−^ → NO pathway after acute BRJ consumption might not affect sympathetic vasoconstriction in either the artery or vein during exercise.

As mentioned above, we found no effect of BRJ on the venous vascular response in the non-exercising limb to exercise. Although we have no conclusive explanation, there are several possibilities for this result. First, in our study, the exogenous production of NO by BRJ was perhaps observed significantly because of increased NO_3_^−^ by 574 ± 120 µM 2 h after intake of BRJ, although it may not contribute to the attenuation of venous vascular tone (relaxation of smooth muscle) during exercise for the following reason. Our study participants were healthy young adults who were expected to have normal endothelial function. This meant that the endogenous NO production may have been sufficient to control the venous vascular tone. Indeed, in previous studies investigating the effect of BRJ on arterial endothelial function, flow-mediated dilation (FMD) was improved in participants with impaired endothelial function, including older adults [27,34], overweight or obese men [35], and patients with hypertension [2], but the FMD was very slightly increased [7] or unchanged [4,36] in healthy humans. Second, the inhibition of efferent sympathetic nerve activity induced by the NO increase with BRJ intake might not perhaps be observed in our study because plasma catecholamines during exercise have been reported to not be altered by acute intake of BRJ when compared with placebo [37]. Because the available scientific evidence is limited, we need to further investigate the effects of vasodilator action and inhibition of sympathetic nerve activity induced by the BRJ supplementation-related increase in NO on the venous vascular responses to exercise.

SBP, DBP, and MAP at pre-exercise and during exercise did not differ between CON and BRJ (Table 1). In addition, the degrees of the increases in these parameters with exercise were also similar for CON and BRJ (Figure 2A–D). Consistent with the present findings, previous studies have also reported no effects of acute dietary NO_3_^−^ supplementation on resting BP [38,39] and BP during exercise [9,37] in young normotensive individuals. In contrast, some studies reported a reduction in BP following acute dietary NO_3_^−^ supplementation with BRJ [1,3,4,40,41]. Similarly, there is no consensus on the hypotensive effect of BRJ. Some possibilities may be considered as to why the effect of dietary NO_3_^−^ supplementation on BP differed among studies. First, the individual response to dietary NO_3_^−^ supplementation varies widely [41,42]. Second, eating habits might differ among participants. For example, “traditional” foods found in a Japanese diet appear to be high in NO_3_^−^ [24]. Because chronic intake of a NO_3_^−^-rich diet might mitigate the ability of dietary NO_3_^−^ supplementation to increase the levels of biomarkers of NO synthesis [43], chronic exposure to a NO_3_^−^-rich Japanese diet may have contributed to the lack of a significant effect on BP, even though the participants were requested to refrain from consuming these foods throughout the study period. Finally, aerobic fitness levels might have varied among participants because NO synthase has been reported to be increased by physical activity [44].

Interestingly, in the individual data, a smaller decrease in calf venous volume during exercise with BRJ intake rather than CON was significantly related to a lower MAP elevation with exercise under BRJ conditions (Figure 3D). In other words, if the venoconstriction in the calf during exercise is attenuated by BRJ, the MAP pressor response to exercise may also be lower and in line with the degree of venoconstriction attenuation. In our previous study, in the individual data, a greater increased venous compliance with BRJ tended to be associated with a lower resting BP under BRJ conditions [19]. Based on the present and previous findings, the mechanism for the hypotensive effect of BRJ might be partly attributed to the control of venous vascular tone. Elevated venous stiffness appears to at least somewhat be a factor in the pathogenesis of hypertension [22]. Thus, it is very important to understand the influence of nutritional component(s) with hypotensive effects on venous vascular control, and our present results may boost the development of interventions to improve and maintain vascular health.

This study has several limitations. First, the venous vessel is modulated by an active factor that indicates activation of the sympathetic nerve [12,14,15,16] and by a passive factor that shows the change in volume-flow dependence [45]. The decrease in limb volume has been reported to reflect the venoconstriction or the elevation of venous tone when the intravascular pressure of the large vein in the limb is maintained at a constant level [15,16]. In addition, the cuff inflation pressure can be considered equivalent to the intravascular pressure of the large vein in the limb [28]. Considering these findings, we believe that the decreases in venous volume in the forearm and calf under cuff pressure of a constant subdiastolic BP of 30–40 mmHg in the present study were caused by sympathetic activation so that this decreased venous volume in limbs reflects the venoconstriction. Second, we did not quantify NO_2_^−^ concentrations. This is important because the conversion of NO_3_^−^ to NO_2_^−^ is necessary for biological effects to occur [4]. However, because there is evidence that a significant increase in the plasma NO_3_^−^ level is accompanied by an increase in NO_2_^−^ concentration in healthy young adults [40,46,47], we believe that the plasma NO_2_^−^ concentration was also elevated after consumption of BRJ in our study. Finally, because the small number of limited participants might have introduced a degree of sampling bias, several variables can interfere with the results and conclusions inferred. Thus, our results cannot be generalized, and the conclusion of our study is applicable to a limited extent to the healthy young adults who participated in this study.

## 5. Conclusions

In this study, we investigated the effect of acute dietary NO_3_^−^ supplementation with BRJ on the venous vascular response and circulatory responses to exercise. Our findings suggest that acute BRJ supplementation leads to an increase in the plasma NO_3_^−^ concentration but does not change venoconstriction in the non-exercising limbs or the MAP response to static exercise in young healthy adults. In addition, although BRJ does not cause a decrease in venoconstriction in all subjects, when venoconstriction during exercise is attenuated after BRJ ingestion, the MAP pressor response to exercise may be reduced to a degree dependent on the attenuation of venoconstriction.

## Figures and Tables

**Figure 1 nutrients-14-04464-f001:**
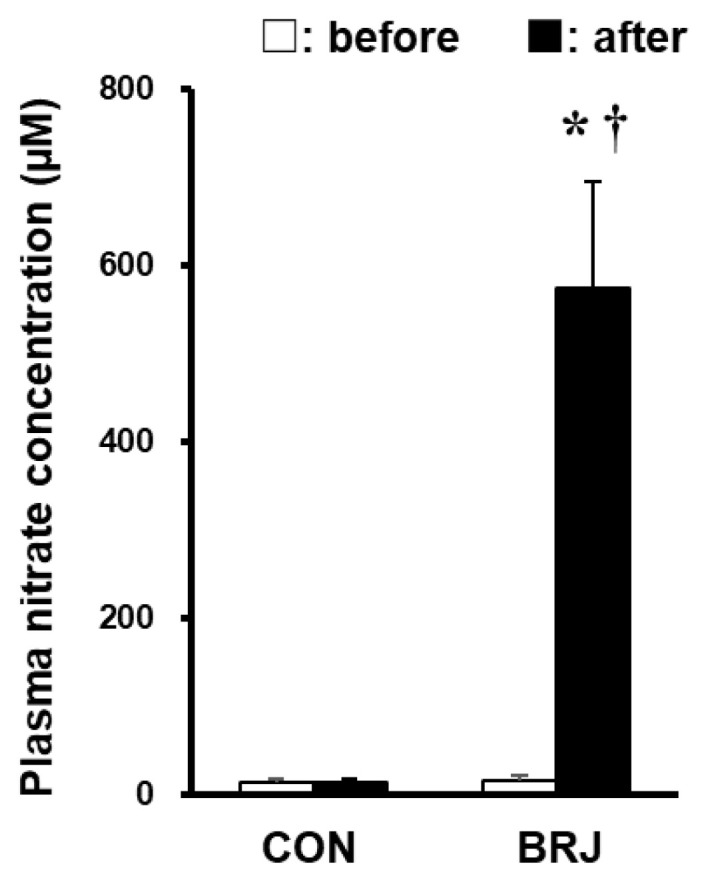
Plasma nitrate concentration before and 2h after CON and BRJ consumption. Data are shown as the mean ± standard deviation. CON: NO_3_^−^-depleted prune juice; BRJ: beetroot juice. * *p* < 0.05, significant difference between before and after. † *p* < 0.05, significant difference between CON and BRJ. Data are expressed as values for n = 16 (10 men and 6 women).

**Figure 2 nutrients-14-04464-f002:**
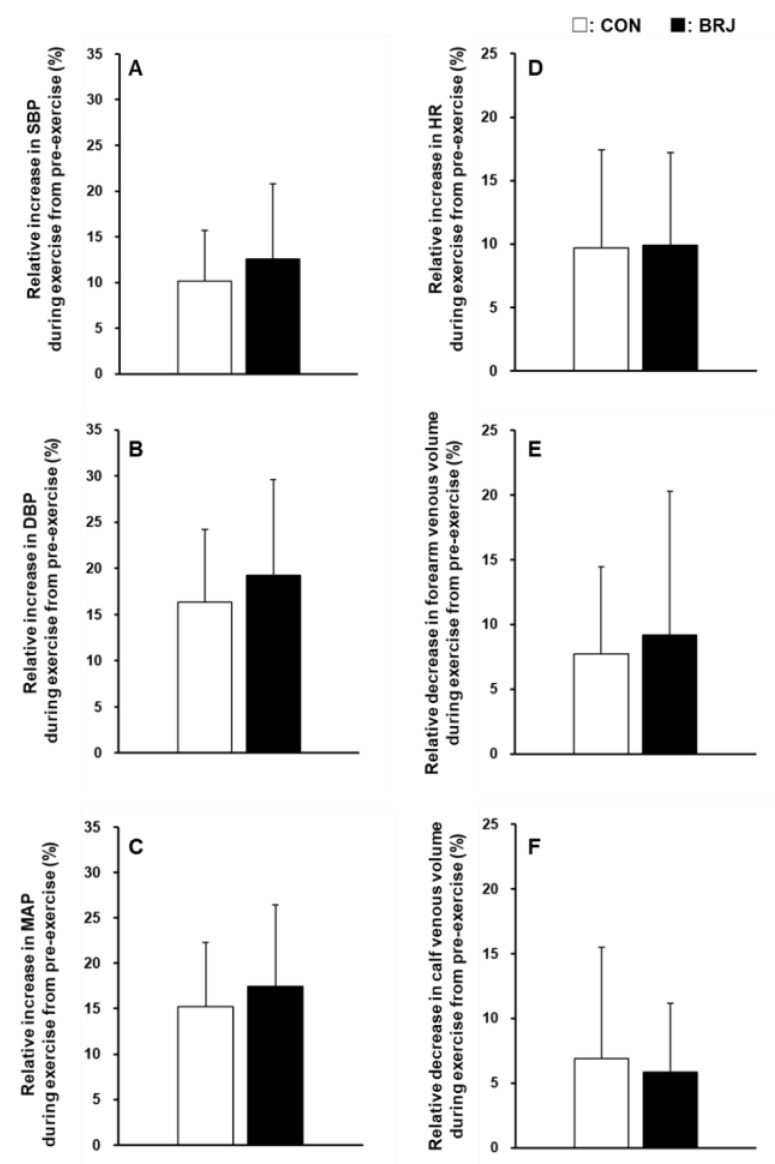
Relative changes in circulatory parameters and venous volume in the non-exercising limbs with static handgrip exercise in CON and BRJ. Percentage increases in the systolic blood pressure (SBP; (**A**)), diastolic blood pressure (DBP; (**B**)), mean arterial pressure (MAP; (**C**)), and heart rate (HR, (**D**)) and percentage decreases in venous volume in the forearm (**E**) and calf (**F**) during exercise from pre-exercise. Data are shown as the mean ± standard deviation. CON: NO_3_^−^-depleted prune juice; BRJ: beetroot juice. Data are expressed as values for n = 16 (10 men and 6 women).

**Figure 3 nutrients-14-04464-f003:**
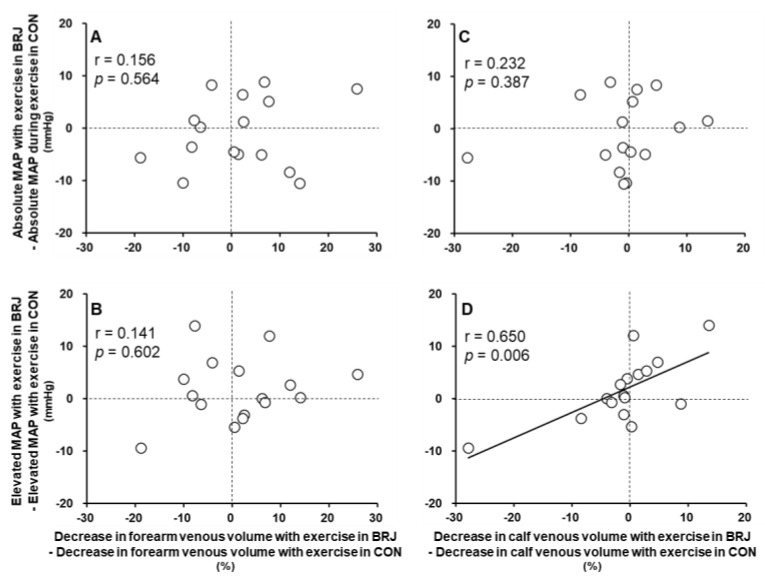
Relationship between the mean arterial pressure response and venous vascular response to exercise. Changes in the absolute mean arterial pressure (MAP) at 2 min of exercise (**A**,**C**) and elevated MAP with exercise (**B**,**D**) from BRJ to CON were plotted against changes in decreased venous volume in the non-exercising forearm and calf with exercise from BRJ to CON. Associations were determined with the use of a Spearman’s rank correlation coefficient assessment. Data are expressed as values for n = 16 (10 men and 6 women).

**Table 1 nutrients-14-04464-t001:** Changes in circulatory parameters and venous volume in non-exercising limbs with static handgrip exercise in CON and BRJ.

	Pre-Exercise	During Exercise at 2 min	ANOVA *p*
SBP, mmHg			
CON	126 ± 13	139 ± 15 *	Condition: 0.346 Time: 0.001 Condition × Time: 0.346
BRJ	123 ± 13	139 ± 15 *	
DBP, mmHg			
CON	75 ± 6	87 ± 7 *	Condition: 0.445 Time: 0.001 Condition × Time: 0.203
BRJ	73 ± 7	86 ± 11 *	
MAP, mmHg			
CON	95 ± 8	110 ± 11 *	Condition: 0.334 Time: 0.001 Condition × Time: 0.283
BRJ	93 ± 9	109 ± 14 *	
HR, bpm			
CON	65 ± 13	71 ± 12 *	Condition: 0.339 Time: 0.001 Condition × Time: 0.767
BRJ	64 ± 11	70 ± 12 *	
Forearm venous volume, mL/dL of tissue			
CON	1.63 ± 0.53	1.52 ± 0.54 *	Condition: 0.682 Time: 0.001 Condition × Time: 0.369
BRJ	1.69 ± 0.40	1.54 ± 0.40 *	
Calf venous volume, mL/dL of tissue			
CON	2.02 ± 0.76	1.90 ± 0.79 *	Condition: 0.502 Time: 0.001 Condition × Time: 0.607
BRJ	2.11 ± 0.70	2.01 ± 0.72 *	

Values are mean ± standard deviation. SBP: systolic blood pressure, DBP: diastolic blood pressure, MAP: mean arterial pressure, HR: heart rate. * *p* < 0.05, significant difference between pre-exercise and during exercise at 2 min. Data are expressed as values for n = 16 (10 men, 6 women).

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
