# Peer review of "Effect of Acute Dietary Nitrate Supplementation on the Venous Vascular Response to Static Exercise in Healthy Young Adults"

_nutrients, 2022, doi:10.3390/nu14214464_

Round 1
Reviewer 1 Report
In the manuscript by Oue et al, entitled "Effect of acute dietary nitrate supplementation on the venous vascular response to static exercise in healthy young adults", the influence of NO3- supplementation on vasoconstriction was evaluated.
Generally speaking, I think manuscript writing can be simplified. The information is presented in a somewhat confusing way, namely in the introduction. In which, the presentation of ideas is not done in a chained way and is easy to read and understand. Furthermore, the question that raises the most doubts and concerns for me is whether you consider that a sample of only 16 individuals without gender homogeneity is representative.
Following Nutrients guidelines, Remove ")" in authors' affiliations.
The first sentence of the abstract is too long, try to keep it simple.
In the procedure, they must place information on all the equipment used, for example, line 157.
In the results, pay attention to the number of significant figures.
The p-value symbol is a lowercase P.
Is recommended that the bibliographic references have less than 10 years. Verified.
I consider that a non-homogeneous gender sample of only 16 individuals is not sufficient to draw representative conclusions regarding a supplementation study. Several variables can interfere with the results and conclusions inferred.
Author Response
Response to Reviewer 1 Comments
Comment 1: In the manuscript by Oue et al, entitled "Effect of acute dietary nitrate supplementation
on the venous vascular response to static exercise in healthy young adults", the influence of NO3-
supplementation on vasoconstriction was evaluated.
Response 1: We wish to thank the Reviewer #1 for the time and attention given to our paper. We
have carefully revised our manuscript according to the comments, suggestions, and concerns raised
by the Reviewer #1. In doing so, we believe that our manuscript has been substantially improved.
Please note that the revised parts have been written in red in the revised manuscript. In addition, we
have provided point-by-point answers to the comments as below.
Comment 2: Generally speaking, I think manuscript writing can be simplified. The information is
presented in a somewhat confusing way, namely in the introduction. In which, the presentation of
ideas is not done in a chained way and is easy to read and understand. Furthermore, the question
that raises the most doubts and concerns for me is whether you consider that a sample of only 16
individuals without gender homogeneity is representative.
Response 2: Thank you for your constructive comments. We have polished the Introduction and have
simplified the overall text to make it easier to read and understand. (P1 L27 ‒ P2 L54)
In addition, as the comment of Reviewer #1, since the data in this study was obtained in a small
number of limited individuals, we consider that our results may not necessarily be applicable to all
population. Thus, we have added this issue to the Limitation as following. (P10 L327‒L331)
“Finally, because the small number of limited participants might have introduced a degree of
sampling bias, several variables can interfere with the results and conclusions inferred. Thus, our
results cannot be generalized, and the conclusion in our study is applicable to a limited extent to the
healthy young adults who participated in this study.”
Comment 3: Following Nutrients guidelines, Remove ")" in authors' affiliations.
Response 3: “)” in authors’ affiliations has been removed. (P1 L4‒L6)
Comment 4: The first sentence of the abstract is too long, try to keep it simple.
Response 4: We have modified the first sentence of the abstract as following. (P1 L10‒L11)
“The purpose of this study was to test the hypothesis that acute intake of inorganic nitrate (NO3−)
supplementation would attenuate the venoconstriction and pressor response to exercise.”
Comment 5: In the procedure, they must place information on all the equipment used, for example,
line 157.
Response 5: We have added the information of equipment. (P4 L147‒L148)
Comment 6: In the results, pay attention to the number of significant figures.
Response 6: We have noted the number of significant figures, and have corrected the values of NO3−.
(P4 L182‒L183)
2
Comment 7: The p-value symbol is a lowercase P.
Response 7: The p-value symbol has been exchanged from uppercase to lowercase P. (P1 L19, P4
L182, L185, L188, L189, L194, L196, Table 1, Figure 3)
Comment 8: Is recommended that the bibliographic references have less than 10 years. Verified.
Response 8: As the suggestion of Reviewer #1, we have verified our bibliography, and have selected
references to be as close to 10 years or less as possible. Related to this, the reference’s number in the
text have been corrected. However, we have also retained some references older than 10 years that
we judged to be necessary for this paper. We would appreciate your acknowledgement in this regard.
Comment 9: I consider that a non-homogeneous gender sample of only 16 individuals is not sufficient
to draw representative conclusions regarding a supplementation study. Several variables can
interfere with the results and conclusions inferred.
Response 9: Thank you very much for your constructive comment. As we have also responded in
Point 2 mentioned above, because the small number of limeted individuals might have introduced a
degree of sampling bias, we think that our results cannot be generalized. We consider that the
conclusion in our study is applicable to a limited extent to the healthy young adults who participated
in this study. Thus, this point has been added to Limitation. (P10 L327‒L331)

Reviewer 2 Report
This manuscript discusses the role of nitrates as a dietary supplement as a response to static exercise in healthy adults. The goal and the experiments of the manuscript aligns with the theme of the paper. I have a few minor comments on the paper.
1) The authors have attributed the rise of nitrates in their study due to the BRJ, however, they have not shown a figure with statistics of how much that increase is relative to when no BRJ was ingested. A figure would be much easier to understand these changes.
2) The authors have mentioned that the local diet which may contain nitrates may askew their results. Did the authors check for endogenous levels of nitrate before the start of the experiment? This could have some kind of effect on the study
3) While the authors have attributed the non-effect of nitrates to low levels after ingestion of BRJ, is there any study that has investigated for the bioavailability of nitrates from BRJ as a supplement? How quickly or how well do that enter tissues to exert an effect?
Author Response
Comment 1: This manuscript discusses the role of nitrates as a dietary supplement as a response to
static exercise in healthy adults. The goal and the experiments of the manuscript aligns with the
theme of the paper. I have a few minor comments on the paper.
Response 1: We wish to thank the Reviewer #2 for the time and attention given to our paper. We
have carefully revised our manuscript according to the comments, suggestions, and concerns raised
by the Reviewer #2. In doing so, we believe that our manuscript has been substantially improved.
Please note that the revised parts have been written in red in the revised manuscript. In addition, we
have provided point-by-point answers to the comments as below.
Comment 2: 1) The authors have attributed the rise of nitrates in their study due to the BRJ, however,
they have not shown a figure with statistics of how much that increase is relative to when no BRJ was
ingested. A figure would be much easier to understand these changes.
Response 2: Thank you for your comment. As the suggestion of Reviewer #2, we have added the
figure in Results. (P4 L180‒L181, P5 L208‒L211)
Comment 3: 2) The authors have mentioned that the local diet which may contain nitrates may askew
their results. Did the authors check for endogenous levels of nitrate before the start of the experiment?
This could have some kind of effect on the study.
Response 3: We did not check for endogenous levels of nitrate before the start of the experiment.
However, we assessed the plasma nitrate concentration before intake of CON and BRJ (CON: 14 ± 3
µM, BRJ: 15 ± 6 µM), which seemed to be slightly smaller than the previous findings (approximately
20~115 µM; Kim JK et al. Appl Physiol Nutr Metab 40: 122-128, 2015, Stanaway L et al. Nutrients 11:
1683, 2019). As the comment of Reviewer #2, there is possibility that the different endogenous levels
of nitrate between our and other studies were able to have some kind of effect on our study.
Comment 4: 3) While the authors have attributed the non-effect of nitrates to low levels after
ingestion of BRJ, is there any study that has investigated for the bioavailability of nitrates from BRJ
as a supplement? How quickly or how well do that enter tissues to exert an effect?
Response 4: Wylie LJ et al. (J Appl Physiol 115: 325-336, 2013) reported the plasma nitrate
pharmacokinetics after ingestion of three different quantities of BRJ, using the dose of 70, 140, or 280
ml of BRJ corresponding on ~4.2, ~8.4 and ~16.8 mM of nitrate. They reported that the plasma nitrate
concentration was elevated remarkably within 1 h after BRJ consuming in dependent on dose, and
the pronounced increase in plasma nitrate was maintained at least until 4 h after BRJ ingestion. In
addition, in the case of 140 ml of BRJ which was the same amount as in our study, the plasma nitrate
concentration was approximately 250~300 µM 2h and 4h after BRJ intake and the significant decrease
in blood pressure was observed. Thus, it seems to exert an effect about 2 h after BRJ intake on BP at
least at plasma nitrate concentration of 250~300 µM. In our study, plasma nitrate concentration was
elevated by 574 ± 120 µM 2 h after intake of BRJ. This value was approximately twice that of Wylie et
al.. This indicates that plasma nitrate level in our study was not necessarily low, and although the
NO was produced by BRJ intake sufficiently, it may perhaps not contribute to the attenuation of
venous vascular tone. Based on these considerations, because the interpretation for non-effect of BRJ
2
on venous vascular response in the original paper could not be expressed appropriately, we have
corrected some sentences in Discussion as following. (P9 L266‒L269)
“First, in our study, the exogenous production of NO by BRJ was perhaps observed significantly
because of increased NO3− by 574 ± 120 µM 2 h after intake of BRJ, although it may not contribute to
the attenuation of venous vascular tone (relaxation of smooth muscle) during exercise for the
following reason.
